# The Impact of the U.S. Macroeconomic Variables on the CBOE VIX Index

Akhilesh Prasad [1,*], Priti Bakhshi [1] 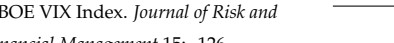 and Arumugam Seetharaman [2]

1   SP Jain School of Global Management, Mumbai 400070, India; priti.bakhshi@spjain.org
2   SP Jain School of Global Management, Hyderabad Road, Singapore 119579, Singapore; seetha.raman@spjain.org
*   Correspondence: akhilesh.dj20dba012@spjain.org

**Abstract:** The purpose of this study is to find the influence of various macroeconomic factors on the volatility index, as macroeconomic factors affect stock market volatility, resulting in an impact on the VIX Index, representing the risk in the stock market. To estimate the significance and importance of the U.S. macroeconomic variables on stock market volatility and risk, classification problems from machine learning are constructed to predict the daily and weekly trends of the VIX Index. Data from May 2007 to December 2021 is considered for analysis. The selected models are trained with twenty-four daily features and twenty-four plus nine weekly features. The outcomes suggest that the decisions made by the Light GBM and XG Boost on ranking features can be significantly accepted over logistic regression. It is found from the results that economic policy uncertainty indices, gold price, the USD Index, and crude oil are signified as strong predictors. The Financial Stress Index, initial claims, M2, TED spread, Fed rate, and credit spread are also strong predictors, while various yields on fixed income securities make a little less impact on the VIX Index. The TED spread, Financial Stress Index, and Equity Market Volatility (Infectious Disease Tracker) are positively associated with the VIX.

**Keywords:** machine learning; feature importance; volatility; Financial Stress Index; money supply; equity market; risk

## 1. Introduction

The CBOE VIX Index is a short-term measure of real-time risk in the stock market and is viewed as a fear index. The day-to-day movements in the VIX Index indicate how the market's perceptions fluctuate over time, and it is an important tool for risk management in the capital market. The movements of the VIX Index from day to day are of interest, not only as a good check on the shifting market perceptions of risk, but also for volatility trading, using options strategies, or VIX futures. Some researchers (Carr 2017; Onan et al. 2014; Sarwar 2012) believe the VIX acts as a fear index or a market perception of risk, while others (Bantwa 2017; Chandra and Thenmozhi 2015) propose risk handling and portfolio diversification.

The VIX Index was initially created by the Chicago Board of Options Exchange for options written on the S&P 100 Index, and later it was shifted to the S&P 500 Index. Since 2003, the VIX Index is the implied volatility of the options written on the S&P 500 Index. Further details including calculation methodology can be found on its White Paper (CBOE VIX White Paper 2021). Since the index is based on observed prices, it provides a market-based appraisal of the riskiness of stocks.

Some researchers (Chaudhary and Bakhshi 2021; Su et al. 2019; Olweny and Omondi 2011; Vähämaa and Äijö 2011; Le et al. 2019) have noted that the macroeconomic factors impact the volatility of the stock market, while others (Okech and Mugambi 2016; Jain and Biswal 2016; Hibbert et al. 2011; Loncarski and Szilagyi 2012) have stated that macroeconomic indicators influence the return on stocks or the stock market in some way.

Considering such an important revelation, the main motivation behind this study is to examine the role of macroeconomic variables on the risk contribution to the stock market, unlike most of the studies on returns where risk is measured by the VIX Index. Though most of the previous studies based on macroeconomic variables examined correlation tests and applied linear regression and hypothesis testing, this study applied a recently developed machine learning technique called the classification technique on daily and weekly data to study the impact of macroeconomic variables on the VIX Index, because the VIX Index is assumed to be the most important measure of the risk of the U.S. stock market. To examine the importance and significance of feature variables derived from daily and weekly macroeconomic variables, the classification techniques called Light GBM (Ke et al. 2017), XG Boost (Chen and Guestrin 2016, August), and logistic regression were applied for predicting the day-to-day and week-to-week upward and downward movements in the VIX Index. Once the optimality of the models was achieved after training and validation, the ranked feature variables were captured, and thereafter the models were asked to predict the VIX target's labels on the testing dataset, though the prediction is not the prime focus of this study.

The remainder of this document is organized as follows: Section 2 discusses past research consisting of macroeconomic variables affecting the stock market, Section 3 lays out the research design and the methodology used, Section 4 displays the findings and their interpretation, and, lastly, Section 5 is the conclusion, with implications and future scope.

## 2. Literature Review

Using the VAR (vector autoregressive) and the EGARCH (exponential general autoregressive conditional heteroskedastic) models, Rizwan and Khan (2007) inspected the importance of domestic and global macroeconomic determinants on the stock returns volatility and found that domestic macroeconomic factors have different levels of significance in describing the association between volatility in the stock market and stock returns. However, the two global factors, the 6-month LIBOR rate, and the MSCI World Index describe the stock returns.

Prasad and Seetharaman (2021) highlighted the importance of machine learning techniques on the trading of financial securities, while Zhong and Enke (2017), by incorporating 60 financial and economic variables that were processed through a thorough data mining process, built classification models, ANNs (artificial neural networks) and logistic regression, to foretell the daily trend of the S&P 500 Index. Additionally, Milosevic (2016) applied various machine learning classification algorithms by considering annual fundamental indicators as features taken from the Bloomberg terminal to evaluate the future price of equity over the long horizon. The author assumed that if the stock price increases by 10%, it is considered "Good", otherwise it is "Bad". "Good" and "Bad" are taken as the target variables.

Using EGARCH and TGARCH, Olweny and Omondi (2011) studied the influence of macroeconomic factors: the inflation rate, forex rate, and variation in the interest rate on the monthly volatility of stock returns at the Nairobi Securities Exchange, and noted that, though the inflation rate, forex rate, and interest rate impacted the stock volatility, the forex rate had a greater impact on the stock return volatility.

To study the ripple effect of U.S. economic uncertainty on volatility in the stock market, Su et al. (2019) applied a bivariate GARCH-MIDAS model and computed its volatility impact on six industrialized and three advanced market countries. Considering U.S. uncertainty indices, FU (financial uncertainty), EPU (economic policy uncertainty), and NVIX (news-implied uncertainty), the results depict that FU does not properly forecast the long-term volatility in the stock market, EPU is positively connected with the volatility in the stock markets of the industrialized countries, and NVIX is more important in the prediction of market volatility; that is, a greater NVIX leads to lesser volatility. Additionally, Hasan et al. (2020) investigated the degree to which the ripple effect of uncertainty among global stock markets is powered by the cross-country association of EPU and applied

data on stock market uncertainty and EPU indices for 13 nations for the period from January 2011 to December 2018. The study found that the EPU association between any two nations remarkably powers the association of uncertainty between their stock markets. Such linkage stands for short- and long-term uncertainty. Le et al. (2019) studied the macroeconomic factors affecting the volatility of stock indices to overcome shortcomings in the Vietnam stock market. Results indicate that, for a small period, the stock market index is causally associated with the M2 money supply, interest rates, and oil prices. However, for a longer duration, the stock market index of Vietnam is affected by the money supply, interest rate, oil price, SJC gold price, and exchange rate. Moreover, based on a survey-based study using a correlation study and econometric modeling, Markowski and Keller (2020) analyzed the influence of 80 macroeconomic variables on the level of the VIX Index and found that the unemployment rate is the most impactful. Cheng (1995) discovered that there was a positive association between the security returns and the unemployment rate. Park (1997) discovered that growth in the employment rate had the greatest negative impacts on stock returns.

Shaikh and Padhi (2013) examined the influence of macroeconomic announcements from RBI, a central bank in India, on the India VIX Index and revealed from the study that during the pre-announcement period, the India VIX Index increased significantly, while once the announcement was made, the India VIX Index returned to the original level, as the uncertainty was resolved. Furthermore, the result also revealed that the India VIX Index decreased sharply after the planned GDP announcement news but increased sharply on the declaration of monthly inflation rates. The combined impact of the macroeconomic announcements of the employment rate, GDP, monetary policy, and industrial output was discovered to be statistically significant with a negative association.

Vähämaa and Äijö (2011) investigated the impact of Fed's monetary policy on the VIX Index and found that the Fed's policy decisions significantly affected stock market uncertainty. Particularly, following the FOMC meeting decreased the implied volatility. However, the target rate surprises were positively related to market uncertainty.

Grieb et al. (2016) investigated the influence of macroeconomic announcements on the VIX Index and found that a change in the VIX Index was negative on the day when the FOMC and the employment rate were publicly announced, and that during the PPI and CPI announcement day, a decreasing level in the VIX Index was partly explained by seasonal patterns. Gustavsen and Oterhals (2018) built a model based on empirical observation and explored the impact of macroeconomic variables on the VIX Index. This model did not try to explain the effect of different variables on actual volatility. However, the model tried to explain what affected the expectations of the market's volatility over the forthcoming 30 days by measuring the impact of the release of new information about macroeconomic conditions. The results depicted that the market regarded the actions taken by the Federal Open Market Committee (FOMC), as such a decision is very important for its outlook on the economy.

Using quantile regression, Xiao et al. (2019) examined the influence of variations in the implied volatility index of the oil market (OVX) on the variations in the implied volatility index of the Chinese stock market (VXFXI). The authors found that the effects of the variations in the OVX Index on the variations in the VXFXI Index were positive and were more pronounced in the falling markets. Bai and Cai (2021) applied machine learning techniques on 278 economic and financial variables to foretell the daily movement on CBOE VIX Index and found that adaptive boosting achieved an average rate of 57%. Additionally, it was observed that the predictability was mostly contributed by the technical indicators of some constituent stocks of the S&P 500 and the weekly U.S. jobless report. Utilizing the augmented HAR (heterogeneous autoregressive) method with exogenous covariates, Han et al. (2015) investigated the predictability of financial and macroeconomic factors for estimating the VKOSPI Index, an implied volatility index inherited from the options written on the KOSPI 200, and found that few domestic macroeconomic factors could describe the variation in the VKOSPI Index, and that the returns on S&P 500 Index and the VIX Index

played an essential role in forecasting the level of the VKOSPI Index, though the return on the domestic stock market did not forecast the VKOSPI Index.

In view of the concern for the traders during early 2018′s spike in the VIX Index, Canorea (2018) studied the relationship between the VIX Index and industrial metals, and suggested that the CRB commodities index and the VIX Index shared a long-term negative correlation which further improves over the longer horizon. The author further said that base metals and commodities have a positive correlation and concluded that industrial metals and the VIX Index have a negative relationship. To examine the influence of macroeconomic factors on the stock returns of the listed banks in the Nairobi Securities Exchange (NSE), Okech and Mugambi (2016) conducted a statistical analysis and applied OLS (ordinary least squares) regression to determine the regression coefficients. It was estimated from the statistical analysis that the inflation rate, forex rate, and interest rate have a noticeable influence on banking stock returns. However, the GDP had no significant impact at a 5% level of significance. Jain and Biswal (2016) examined the dynamic linkage among the USD/INR forex rate, crude oil prices, gold prices, and the Indian stock market. The result indicated that a decline in crude oil prices and gold prices caused the deterioration in the INR and a decrease in the value of the stock index.

Hibbert et al. (2011) discovered a roughly opposite association between changes in the bond yield spread and the stock return of the issuing firm. Loncarski and Szilagyi (2012) stated a negative association between changes in credit spreads and changes in both the equity index returns and the risk-free short-term interest rate.

During the review of the related past studies, it has been discovered that some studies applied regression techniques while others used different statistical techniques such as hypothesis testing to discover the statistical relationship between the macroeconomic variables and the stock market. Additionally, some other studies (Chaudhary et al. 2020a, 2020b) focused on analyzing volatility using traditional statistical methods and by incorporating financial variables. This confirms that most authors previously examined the linear association on mostly weekly and monthly macroeconomic data in a regression setting in which the target variable is the number. However, it has been seen that previous studies did not look up classification problems for studying the influence of macroeconomic variables on stock market volatility. The classification technique, a machine learning approach, gives the score of the applied feature variables while modeling and predicting the target labels. The score of the feature variables indicates their sensitivity to the target. Therefore, it is important to construct classification problems for analyzing the impact of daily and weekly macroeconomic variables on the stock market volatility index.

## 3. Research Design and Methodology

### 3.1. Data Pre-Processing

To analyze the significance of macroeconomic variables on the prediction of the implied volatility index (VIX), the required daily data are downloaded for the period from May 2007 to December 2021 from various related sources, as listed in Tables 1 and 2, and their features are computed from them, as also mentioned in Tables 1 and 2. The daily data for the VIX Index is downloaded directly from the CBOE portal and does not have any missing values. Because the macro data belong to different departments, it is observed that the macro data is missing on a particular day, but the stock market might be functioning on the same day. In such a situation, the macro data are forward-filled. Moreover, daily macroeconomic variables might suffer from a delay in reporting the issue. To subside this issue, daily macroeconomic variables with delays in reporting issues were not selected in the study and the weekly macroeconomic variables, which are generally free from such issues, were also selected.

**Table 1.** List of daily and weekly features.

| Feature Symbols | Features Description | Source (Symbol) |
|---|---|---|
| C VIX | CBOE implied volatility index | CBOE |
| R Gold Price | Average of gold fixing price (London a.m. and p.m. time) in London Bullion Market, based in U.S. dollars | FRED |
| R Silver Price | Silver fixing price (London noon time) in London Bullion Market, based in U.S. dollars | Quandl |
| 1M T-Yield Curve Rates | Treasury yield curve rates (1-month maturity) | Quandl |
| 4W Bank Discount Rate | Treasury bill rates (4-week bank discount rate) | Quandl |
| Yield Spread | 10-year Treasury constant maturity minus 3-month Treasury constant maturity | FRED |
| BBB Corp OAS | ICE BofA BBB U.S. Corporate Index option-adjusted spread | FRED |
| AAA Corp OAS | ICE BofA AAA U.S. Corporate Index option-adjusted spread | FRED |
| BBB Corp Yield | ICE BofA BBB U.S. Corporate Index effective yield | FRED |
| AAA Corp Yield | ICE BofA AAA U.S. Corporate Index effective yield | FRED |
| Corp OAS | ICE BofA U.S. Corporate Index option-adjusted spread | FRED |
| Corp Yield | ICE BofA U.S. Corporate Index effective yield | FRED |
| C EPU Index | Economic Policy Uncertainty Index for the United States | FRED |
| C EPU Equity Index | Equity market-related Economic Uncertainty Index | FRED |
| C EPU IDT Index | Equity Market Volatility: Infectious Disease Tracker | FRED |
| 5Y Yield Inflation Indexed | Market yield on U.S. Treasury securities at 5-year constant maturity, inflation-indexed | FRED |
| 10Y Yield Inflation Indexed | Market yield on U.S. Treasury securities at 10-year constant maturity, inflation-indexed | FRED |
| TED Spread | Treasury–Eurodollar spread It is the difference between 3-month LIBOR based on U.S. dollars and 3-month Treasury bill rate | FRED |
| 5Y Breakeven Inflation Rate | 5-year breakeven inflation rate It represents a measure of expected inflation derived from 5-year Treasury constant maturity securities and 5-year Treasury inflation-indexed constant maturity securities | FRED |
| 10Y Breakeven Inflation Rate | 10-year breakeven inflation rate It represents a measure of expected inflation derived from 10-year Treasury constant maturity securities and 10-year Treasury inflation-indexed constant maturity securities | FRED |
| Credit Spread | It is the difference between Moody's Seasoned Baa corporate bond yield and Moody's Seasoned Aaa corporate bond yield | FRED |
| Fed Rate | Federal funds effective rate | FRED |
| C OVX | CBOE Crude Oil ETF Volatility Index | Yahoo Finance |
| R USD Index | U.S. Dollar Index | Yahoo Finance |

Notes: Prefix "C" = change in value; Prefix "R" = log return; ICE = Intercontinental Exchange; BofA = Bank of America; ETF = exchange-traded fund; Quandl: https://data.nasdaq.com accessed on 24 February 2021; CBOE: https://cdn.cboe.com/api/global/us_indices/daily_prices/VIX_History.csv accessed on 24 February 2021; FRED: https://fred.stlouisfed.org accessed on 24 February 2021.

**Table 2.** List of additional weekly features.

| Feature Symbols | Features Description | Source (Symbol) |
|---|---|---|
| 30Y Mortgage Rate | 30-year fixed-rate mortgage average in the U.S. | FRED |
| 3m T-Bill 2nd Rate | 3-month Treasury bill secondary market rate | FRED |
| 4w T-Bill 2nd Rate | 4-week Treasury bill secondary market rate | FRED |
| M2 Money Supply | M2 money supply | FRED |
| Financial Stress Index | St. Louis Fed Financial Stress Index | FRED |
| Bank Prime Loan Rate | Bank prime loan rate | FRED |
| R Initial Claims SA | It is a claim filed by individuals to receive unemployment benefits after separation from an employer | FRED |
| R Initial Claims NSA | It is a claim filed by individuals to receive unemployment benefits after separation from an employer | FRED |
| R Crude Oil Price | Crude oil prices: West Texas Intermediate (WTI)—Cushing, Oklahoma | FRED |

Note: SA = seasonally adjusted; NSA = not seasonally adjusted; Prefix "R" = log return.

For some features, in the cases where the log return or first difference needed to be computed, data are forward-filled after computing the log return or first difference. To accommodate the missing data properly, the timestamp of the VIX Index is taken as the base sequence and the data for the other features were forward-filled or, in case of additional data, were deleted after performing an inner join. The weekly data are considered in this study because the weekly data are free from delays in reporting issues.

*3.2. Feature Variables*

The list of feature variables consisting of twenty-three daily and thirty-two macro features, and a feature derived from the VIX itself, has been prepared and is listed in Tables 1 and 2, in which the prefix "R" and "C" of the feature symbols indicate a log return and change (first difference), respectively. Some of the weekly data are prepared from daily series, as stated in Table 1, while some are weekly series downloaded directly from their respective sources, as mentioned in Table 2. The log returns are computed for the features which have trend patterns, while the change in value is computed for the features which have a mean-reverting tendency. The suffix "−1" to "−5" indicates the number of days prior to the current days. For instance, "−1" denotes 1 day prior to the current day's or previous day's value.

*3.3. Target Variables*

The target labels are created from a change in the closing value of the VIX Index. The target variable is taken as 1 when on the next day the VIX is going to be up, and otherwise it is 0. It can be mathematically represented by:

$$y_t = 1 \; when \; Close_t - Close_{t-1} > 0$$

$$y_t = 0 \; otherwise$$

where subscript $t$ indicates the day for daily models and the week for the weekly models, and $y$ is the target label's indication of upward and downward movements.

### 3.4. Construction of the Model

In this study, both the daily and weekly models are considered. For daily models, the previous five days of twenty-four features are considered to capture a wider dynamic for analysis and are fed into the model. By doing so, the model finally has 120 (=24 × 5) predictors. The daily model can be mathematically depicted as:

$$y_t = f(X_{t-1}, \ldots, X_{t-5})$$

where subscript $t$ is time in days and $X$ is the two-dimensional feature matrix.

Similarly, for the weekly models, thirty-three features of the previous week are only fed into the models. The model can be mathematically stated as:

$$y_t = f(X_{t-1})$$

where subscript $t$ is time in weeks.

### 3.5. Description of the Models Used

3.5.1. Logistic Regression

Logistic regression is a special type of classifier which uses the sigmoid function and produces predicted values ranging from 0 to 1. Let us assume the $h_\theta(x)$ to be a hypothetical function which is given by:

$$h_\theta(x) = g\left(\theta^T x\right) = \frac{1}{1 + e^{-\theta^T x}}$$

$$\theta^T x = \begin{bmatrix} \theta_0 & \theta_1 & \ldots & \theta_j \end{bmatrix} \begin{bmatrix} x_0 \\ x_1 \\ \vdots \\ x_j \end{bmatrix}$$

where $\theta$, $x$, and subscript $j$ are the coefficient, predictor variables, and the identity of predictor variables, respectively. During the iteration process, it minimizes the loss, which is the average loss over the training sample. The loss function is given by:

$$L(\theta) = -\frac{1}{M} \sum_{i=1}^{M} y^{(i)} \log\left[h_\theta\left(x^{(i)}\right)\right] + \left[1 - y^{(i)}\right] \log\left[1 - h_\theta\left(x^{(i)}\right)\right]$$

where $y$ is the target label, which is 0 or 1, subscript '$i$' is the instance of the training sample, and $M$ is the number of training samples. Furthermore, the regularization term can be appended to the loss function for protecting against overfitting. Then, the objective function comprises two parts: standard loss and the regularization term.

$$obj(\theta) = L(\theta) + \Omega(\theta)$$

where $L$ is the standard loss function and $\Omega$ is the added regularization term, which can be defined as:

$$\Omega(\theta) = \alpha(L_1) + (1 - \alpha)(L_2)$$

After specifying the expression for $L_1$ and $L_2$, the regularization term becomes:

$$\Omega(\theta) = \lambda \left[ \alpha \sum_{i=1}^{M} |\theta_i| + \frac{1 - \alpha}{2} \sum_{i=1}^{M} \theta_i^2 \right]$$

where $\alpha$ is the mixing parameters and $\lambda$ is the regularization penalty. $L_1$ regularization sets the unimportant variables' coefficients to zero, while in the $L_2$ regularization, the least important variables' coefficients converge towards zero. Regularizations play important

roles in ranking and anticipating the sensitivity of the feature variables. The mix of $L_1$ and $L_2$ is called the elastic net, and the amount of mixing is decided by the mixing parameter $\alpha$.

### 3.5.2. Light Gradient-Boosted Machine (Light GBM) Classifier

This algorithm uses histogram-based algorithms and is developed by Ke et al. (2017). Although most engineering applications have been adopted in gradient boosting techniques, the scalability and efficiency are not yet adequate when the sample size is large and the feature dimension is high. The main motive is that they need to read the complete sample for each predictor variable to measure the information gain for all probable split points; during the process, it takes a lot of time. To address this issue, the authors suggest two innovative methods: GOSS (gradient-based one-side sampling) and EFB (exclusive feature bundling).

### 3.5.3. Extreme Gradient Boosting (XG Boost) Classifier

The XG Boost is an advanced estimator and is based on the gradient-boosted decision trees designed for better execution and model efficiency. It is the most popular and advanced machine learning algorithm. Because of its inherent implementation, it is preferred to any other tree-based model. As a part of the Distributed Machine Learning Community, Tianqi Chen initiated a research project which finally shaped into the XG Boost technique (Chen and Guestrin 2016, August).

In the Light GBM and XG Boost, the reg_alpha and reg_lambda hyperparameters are the L1 and L2 regularizations, respectively, which play an important role in features selection and elimination. Additionally, the importance_type hyperparameter, which takes two values, "split" and "gain", decides the type of feature importance to be filled into feature_importances_. In the case of "split", the result contains the numbers of times the feature is used to split the data across all trees, and in the case of "gain", the result contains the average gains of the features.

While the Light GBM and XG Boost provide feature scores that can be utilized to rank the features, logistic regression provides feature coefficients which help to anticipate the directional relationships. These feature coefficients are not actual, but they can be compared because the features are scaled before being fed into the models.

### 3.6. Performance Measurement

The performance of the classifiers is measured by a set of parameters: accuracy score, precision score, recall score, F1 score, and the Matthews correlation coefficient (MCC), which are described and explained in the studies (Sokolova and Lapalme 2009; Ferri et al. 2009). The MCC is a measure of the correlation between the actual and predicted categorical variables (Matthews 1975; Phi Coefficient 2022). An MCC of zero or less than zero indicates that the classifier is useless. Specifically, the MCC is zero for a random model. While performing the hyperparameters tuning, the accuracy score is maximized in the validation dataset. The accuracy score and the MCC, along with the other parameters, are captured from the testing dataset for judging the performance of the classifiers. The t-test on the MCC is also performed for revealing the statistical significance of the MCC. The working mechanism of the t-test is summarized below:

The t-test is applied on the Matthews correlation coefficient (MCC) to check whether the MCC is statistically significant. Specifically, it confirms whether the predicted value of the classifier is statistically associated with the true value in the testing dataset.

Hypothesis of the *t*-test:

**Null Hypothesis:** $\rho = 0$

This means that the predicted value of the classifier is not statistically associated with the true value.

**Alternate Hypothesis:** $\rho \neq 0$

This means that the predicted value of the classifier is statistically associated with the true value.

This is a two-tailed test, and if the *p*-value is smaller than the significance level, the Null Hypothesis is rejected in the favor of Alternate Hypothesis. A comparatively lesser *p*-value makes a more statistically significant association of the predicted value of the classifier with the true value. Though the standard practice of taking the significance level is 0.05, it is taken as 0.0250 in this study. The degree of freedom is equal to the sample size (the size of the test dataset) minus two. The test statistic is computed as:

$$t = \frac{(r - \rho)\sqrt{n - 2}}{\sqrt{1 - r^2}}$$

where r is the computed MCC. Since the hypothesized value of $\rho$ is zero, the above formula reduces to:

$$t = \frac{r\sqrt{n - 2}}{\sqrt{1 - r^2}}$$

Finally, the *t*-test statistic is converted into its *p*-value accordingly from the t-distribution.

*3.7. Validation Procedure*

The entire dataset is split into training and testing sets. The testing set contains only 100 trading days for daily models and 50 weeks for weekly models, while the rest of the preceding data is the actual training dataset utilized for training the model. Similarly, the size of the validation is 100 days and 50 weeks for the daily and the weekly, respectively. A machine learning model requires periodic hyperparameters tuning because predicting a faraway in the future would not be a good idea. Hence, a fixed and sufficient window of the validation and testing dataset would be advisable. To perform the hyperparameters tuning of the models, a grid search cross-validation along with a 2-fold time-series cross-validation was performed, and the optimal hyperparameters, which are external to the models, were determined and are shown in Table 3 for the daily and Table 4 for the weekly models. The two-fold time-series cross-validation internally splits the training data into two sets of sub-training and validation datasets, which are depicted in Figure 1 for the daily and Figure 2 for the weekly models. Models learn from the sub-training set and compute the accuracy scores from the validation set. The grid search cross-validation picks the value from the list of hyperparameters for which the mean accuracy score is the maximum in the validation dataset. After performing the complete set of validations, the hyperparameters are captured, which are listed in Table 3 for the daily models and Table 4 for the weekly models. The validation results are also displayed in Table 5 for the daily and Table 6 for the weekly models. Split 0 and split 1 are captured from the validations of the two sets of splits and the mean score is the average of split 0 and split 1.

**Table 3.** Hyperparameters of daily models.

| Estimators | Hyperparameters |
|---|---|
| Light GBM | n_estimators = 55, objective = 'binary', importance_type = 'gain', max_depth = 2, num_leaves = 3, learning_rate = 0.8478, subsample_for_bin = 40, subsample = 0.01, colsample_bytree = 0.06, boosting_type = 'gbdt', reg_alpha = 23, reg_lambda = 1.12, min_split_gain = $1 \times 10^{-19}$, min_child_weight = $1 \times 10^{-19}$, min_child_samples = 180 |
| XG Boost | n_estimators = 32, max_depth = 3, learning_rate = 0.077, objective = 'binary:logistic', booster = 'gbtree', tree_method = 'approx', eval_metric = 'logloss', gamma = 11.8, reg_alpha = $1 \times 10^{-14}$, reg_lambda = 1.0, min_child_weight = 19.7, subsample = 0.55, colsample_bytree = 0.75, importance_type = 'gain', scale_pos_weight = 1.0279 |
| Logistic Regression | penalty = 'elasticnet', l1_ratio = 0.95, solver = 'saga', C = 80, max_iter = 10, tol = 0.001 |

**Table 4.** Hyperparameters of weekly models.

| Estimators | Hyperparameters |
|---|---|
| Light GBM | n_estimators = 1020, objective = 'binary', importance_type = 'gain', max_depth = 2, num_leaves = 2, learning_rate = 40, subsample_for_bin = 485, subsample = 0.01, colsample_bytree = 0.37, boosting_type = 'gbdt', reg_alpha = $1 \times 10^{-19}$, reg_lambda = 100,000.0, min_split_gain = $1 \times 10^{-19}$, min_child_weight = $1 \times 10^{-19}$, min_child_samples = 306 |
| XG Boost | n_estimators = 26, max_depth = 2, learning_rate = 0.169, objective = 'binary:logistic', booster = 'gbtree', tree_method = 'hist', eval_metric = 'logloss', gamma = 12.0, reg_alpha = 0.1, reg_lambda = 1.0, min_child_weight = 19.7, subsample = 0.65, colsample_bytree = 0.6, importance_type = 'gain', scale_pos_weight = 0.9587 |
| Logistic Regression | penalty = l2, solver = 'sag', C = 0.00021, max_iter = 5, tol = 0.1, class_weight = 'balanced' |

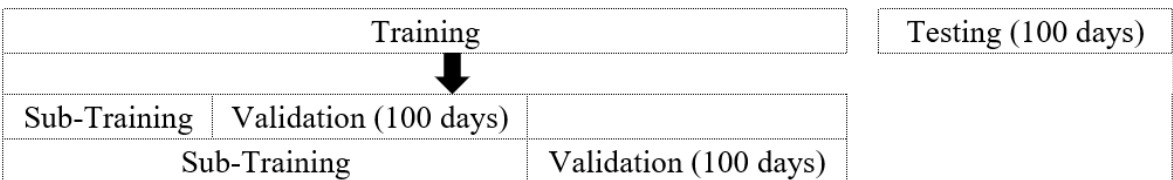

**Figure 1.** The 2-fold time-series cross-validation for daily models.

**Figure 2.** The 2-fold time-series cross-validation for weekly models.

**Table 5.** Validation results from daily models. Numbers are truncated to four decimal places or two decimal places in percent form.

| Accuracy Score | Light GBM | XG Boost | Logistic Regression |
|---|---|---|---|
| split0 validation score | 61.00% | 53.69% | 43.00% |
| split1 validation score | 61.00% | 57.55% | 51.00% |
| mean validation score | 61.00% | 55.62% | 47.00% |

**Table 6.** Validation results from weekly models. Numbers are truncated to four decimal places or two decimal places in percent form.

| Accuracy Score | Light GBM | XG Boost | Logistic Regression |
|---|---|---|---|
| split0 validation score | 62.00% | 58.65% | 56.00% |
| split1 validation score | 62.00% | 56.96% | 56.00% |
| mean validation score | 62.00% | 57.81% | 56.00% |

After performing the validations, the features coefficient and their ranked scores are captured and displayed in the Section 4, and thereafter, the models are tested to predict the labels in the testing dataset. This study is limited to U.S. macroeconomic variables and the U.S. stock market volatility index (CBOE VIX Index).

## 4. Results

To study the significance of the macroeconomic variables on predicting the day-to-day and week-to-week movements of the CBOE VIX Index, the selected set of features are fed into the models and, after performing the hyperparameters tuning as mentioned in the steps in Section 3, the feature coefficients from the logistic regression and the feature scores from the Light GBM and XG Boost are captured and, subsequently, the models are tested to predict the categorical labels of the VIX Index for the testing dataset. During validation, it was observed that the Light GBM ranked twenty-nine daily features and twenty-three weekly features, as depicted in Table A1, and put the scores of the remaining redundant features to zero. The ranked features according to their importance are depicted in Figure 3 for the daily and Figure 4 for the weekly Light GBM classifier. Moreover, XG Boost ranked twenty-two daily features and three weekly features, as displayed in Table A2, and set the scores of the remaining redundant features to zero. The ranked features according to their importance are depicted in Figure 5 for the daily and Figure 6 for the weekly XG Boost classifier. Lastly, logistic regression returned the coefficient of all one hundred and twenty daily features and thirty-three weekly features, as depicted in Tables A3 and A4, respectively. The ranked features according to their significance are stated in Figure 7 for the daily and Figure 8 for the weekly logistic regression. Tables A1–A4 are displayed in Appendix A. The elimination and ranking of the features depend mainly on the regularization parameters, which are required to be set during the hyperparameters tuning for achieving optimal performance in the validation dataset. For the testing dataset, the accuracy scores, the MCC, and the $p$-value are displayed in Table 7, and the classification reports are displayed in Table 8 for the daily and Table 9 for the weekly models. The data displayed in Table 7 are rounded to four decimal places and the data displayed in Tables 8 and 9 are rounded to two decimal places.

**Table 7.** Test scores. Numbers are truncated to four decimal places or two decimal places in percent form.

|  | Frequency | Light GBM | XG Boost | Logistic Regression |
|---|---|---|---|---|
| Accuracy Score | Daily | 62.00% | 62.00% | 52.00% |
| MCC | Daily | 0.2702 | 0.2793 | 0.0162 |
| $p$-value | Daily | 0.0065 | 0.0049 | 0.8733 |
| Accuracy Score | Weekly | 62.00% | 60.00% | 44.00% |
| MCC | Weekly | 0.2522 | 0.2077 | −0.0933 |
| $p$-value | Weekly | 0.0772 | 0.1477 | 0.5192 |

Note: The $p$-value is compared with 2.5% or 0.0250 level of significance.

**Table 8.** Classification reports for daily models. Numbers are truncated to two decimal places.

|  | Logistic Regression | | | XG Boost | | | Light GBM | | | |
|---|---|---|---|---|---|---|---|---|---|---|
|  | precision | recall | F1 score | precision | recall | F1 score | precision | recall | F1 score | support |
| 0 | 0.52 | 0.87 | 0.65 | 0.59 | 0.92 | 0.72 | 0.59 | 0.90 | 0.71 | 52 |
| 1 | 0.50 | 0.15 | 0.23 | 0.78 | 0.29 | 0.42 | 0.75 | 0.31 | 0.44 | 48 |
| macro avg | 0.51 | 0.51 | 0.44 | 0.68 | 0.61 | 0.57 | 0.67 | 0.61 | 0.58 | 100 |
| weighted avg | 0.51 | 0.52 | 0.45 | 0.68 | 0.62 | 0.58 | 0.67 | 0.62 | 0.58 | 100 |

**Table 9.** Classification report for weekly models.

| | Logistic Regression | | | XG Boost | | | Light GBM | | | |
|---|---|---|---|---|---|---|---|---|---|---|
| | precision | recall | F1 score | precision | recall | F1 score | precision | recall | F1 score | support |
| 0 | 0.46 | 0.22 | 0.30 | 0.65 | 0.56 | 0.60 | 0.68 | 0.56 | 0.61 | 27 |
| 1 | 0.43 | 0.70 | 0.53 | 0.56 | 0.65 | 0.60 | 0.57 | 0.70 | 0.63 | 23 |
| macro avg | 0.45 | 0.46 | 0.42 | 0.60 | 0.60 | 0.60 | 0.63 | 0.63 | 0.62 | 50 |
| weighted avg | 0.45 | 0.44 | 0.41 | 0.61 | 0.60 | 0.60 | 0.63 | 0.62 | 0.62 | 50 |

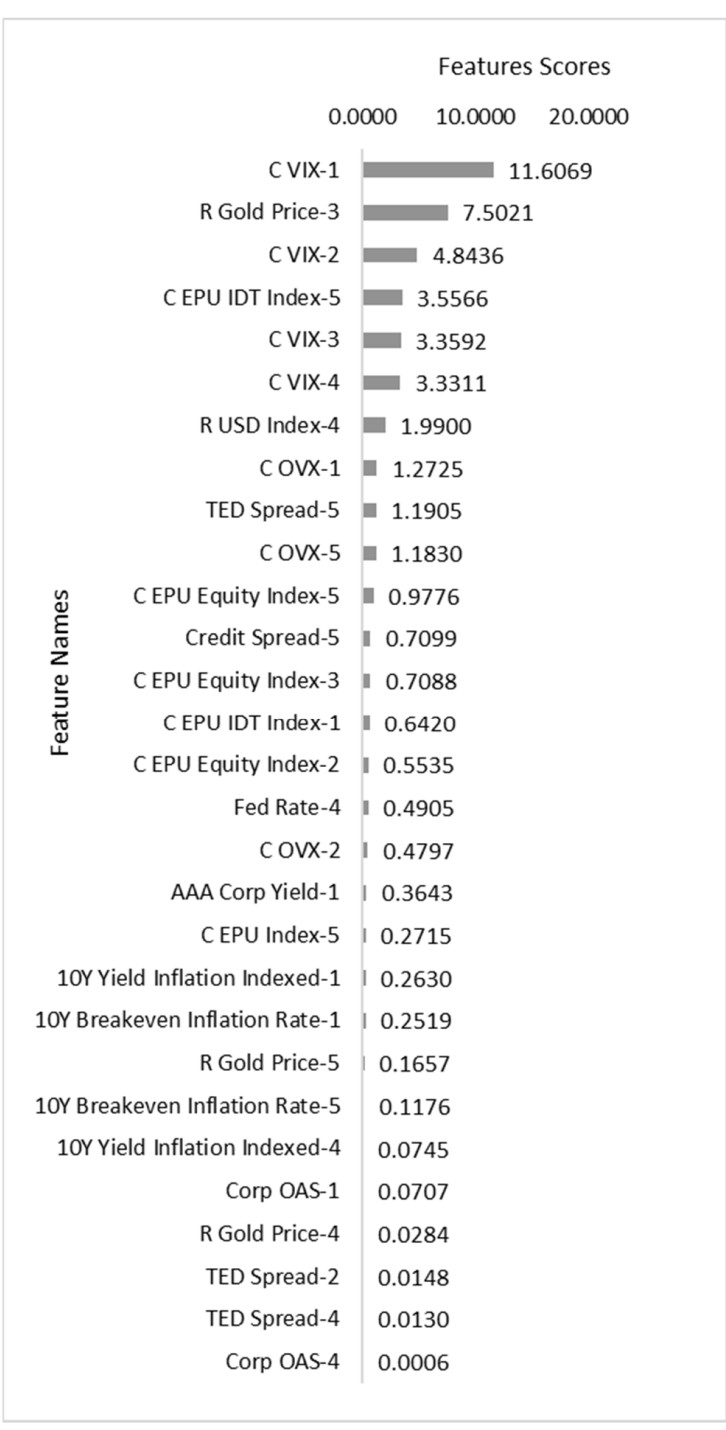

**Figure 3.** Top twenty-nine daily ranked features from Light GBM. The numbers on the right, truncated to four decimal places, are the features' scores.

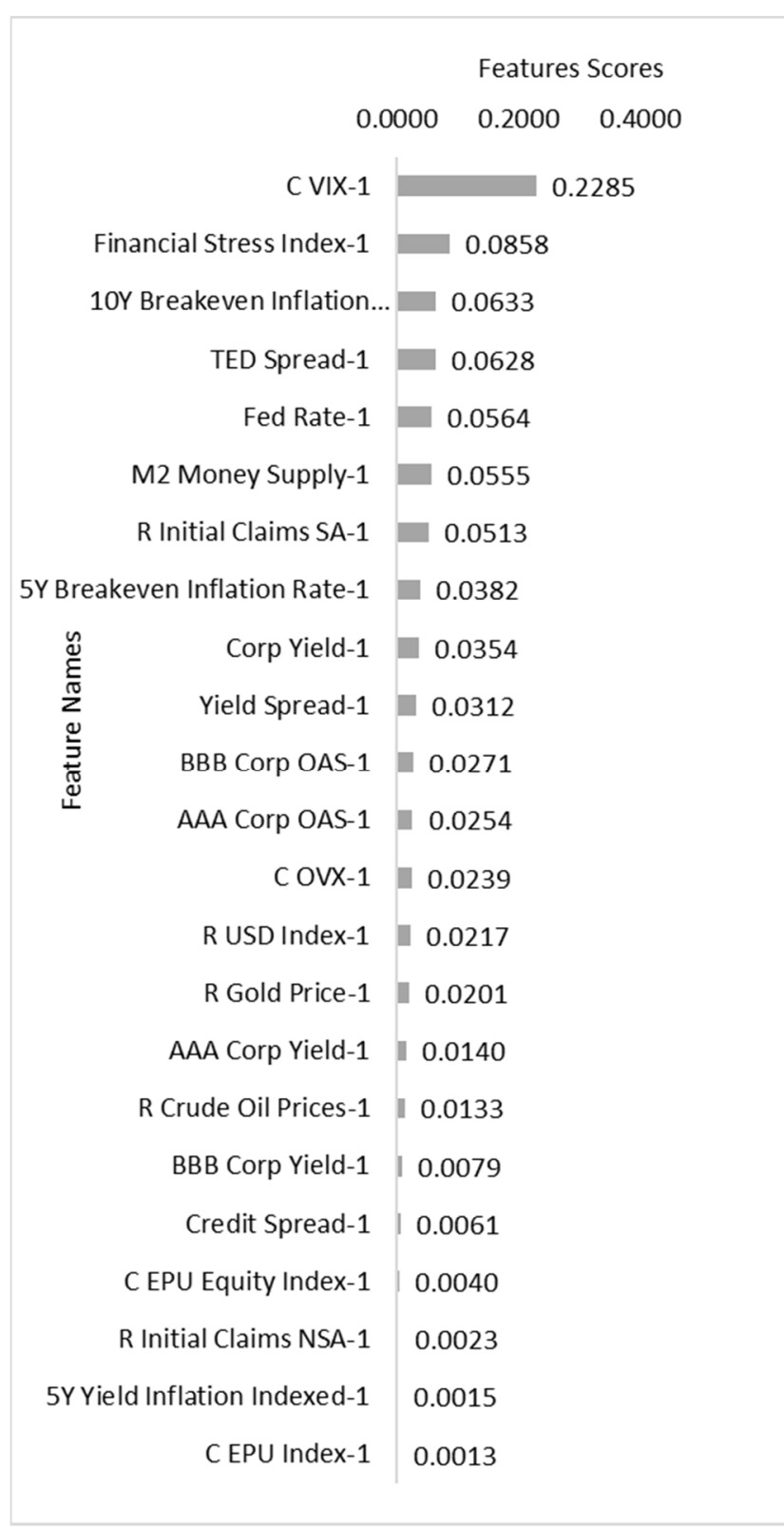

**Figure 4.** Top twenty-three weekly ranked features from Light GBM. The numbers on the right, truncated to four decimal places, are the features' scores.

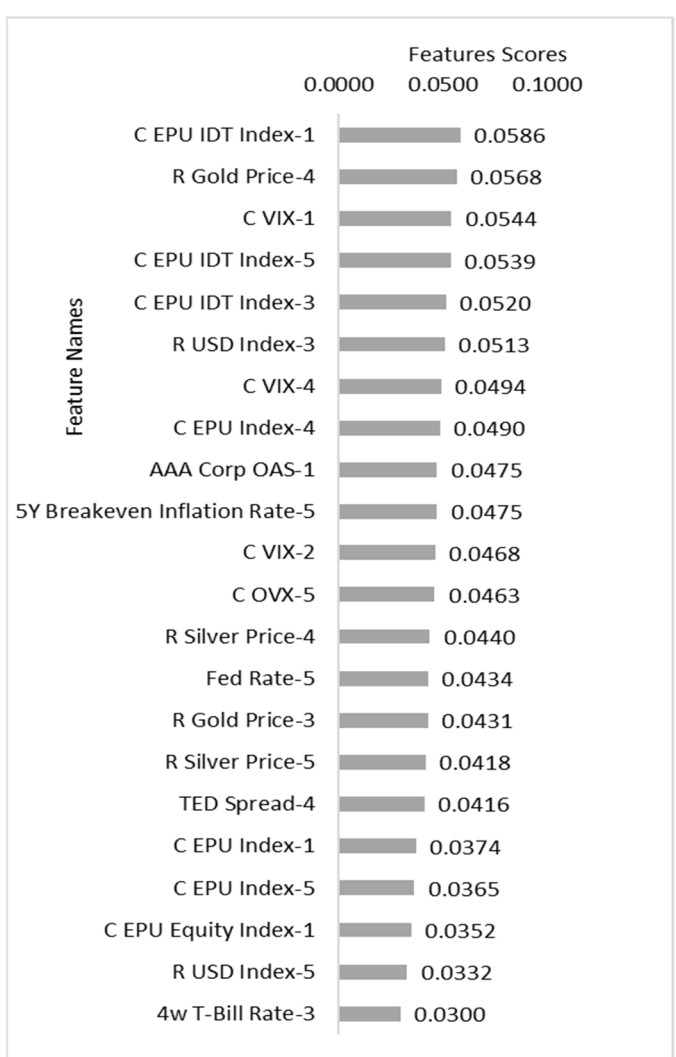

**Figure 5.** Top twenty-two daily ranked features from XG Boost. The numbers on the right, truncated to four decimal places, are the features' scores.

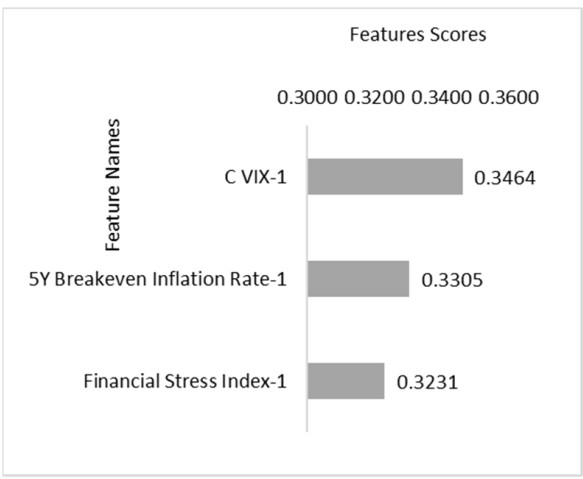

**Figure 6.** Top three weekly ranked features from XG Boost. The numbers on the right, truncated to four decimal places, are the features' scores.

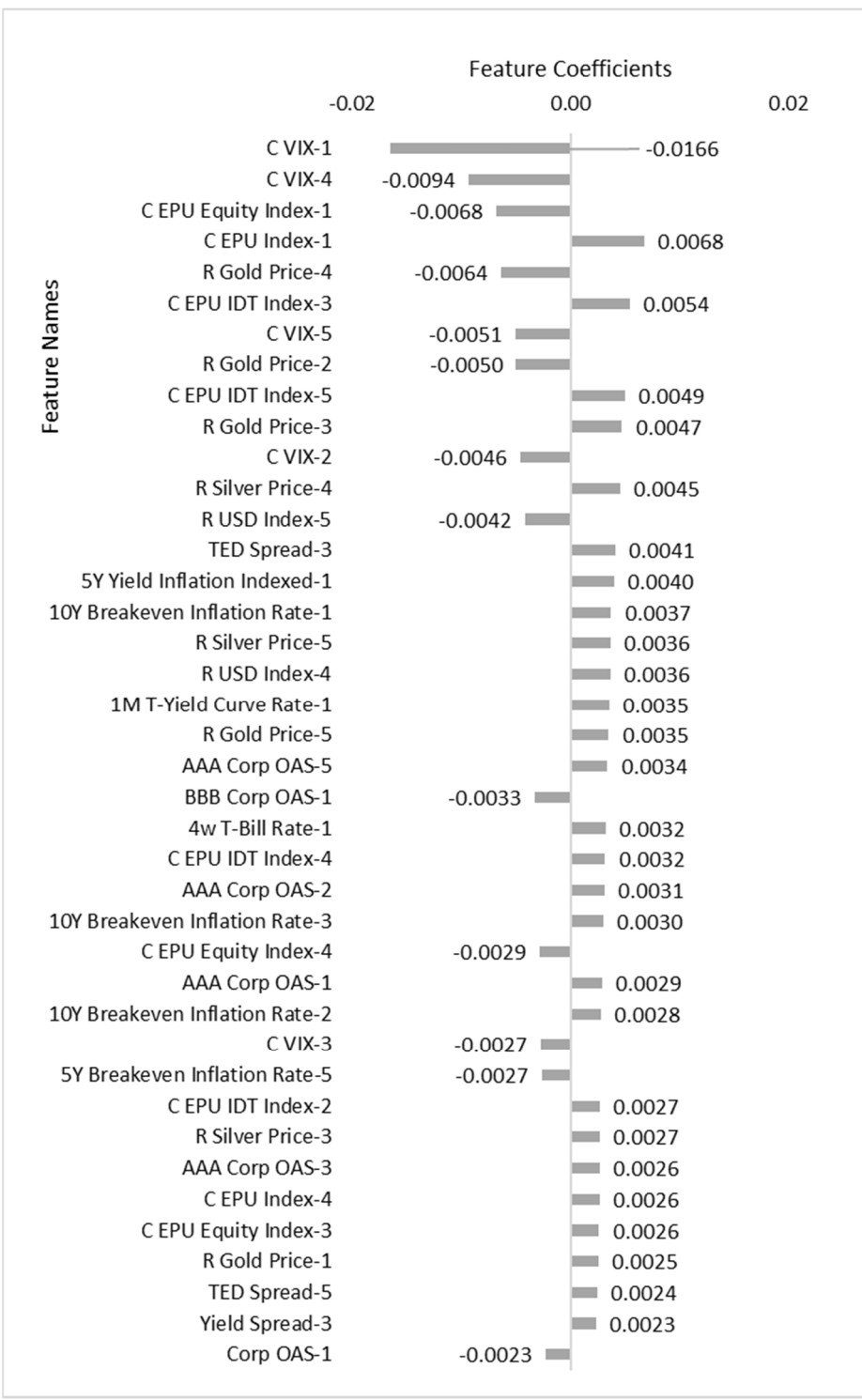

**Figure 7.** Top forty daily ranked features from logistic regression. The displayed numbers, truncated to four decimal places, are the features' coefficients.

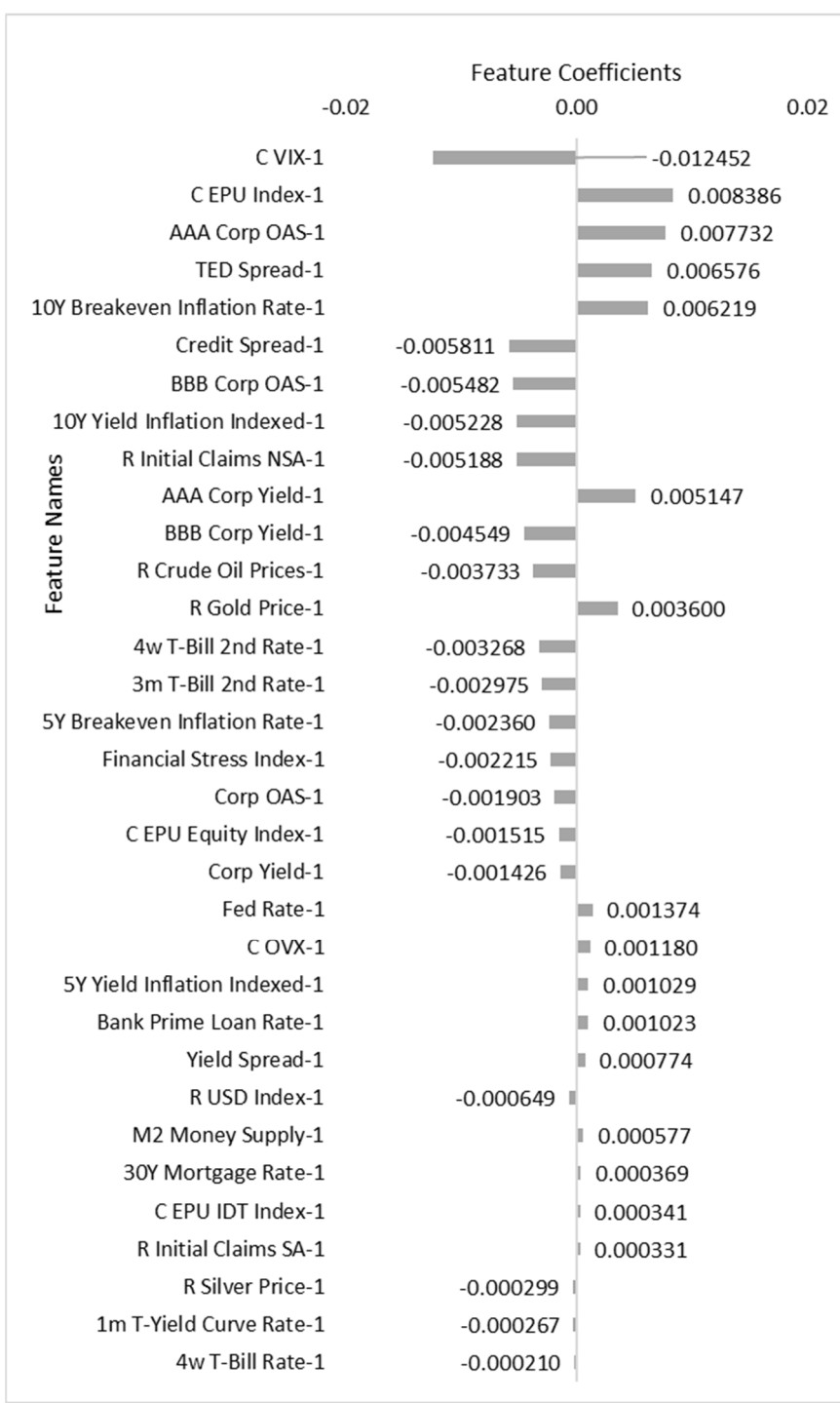

**Figure 8.** Top thirty-three weekly ranked features from logistic regression. The displayed numbers, truncated to four decimal places, are the features' coefficients.

### 4.1. Results from Daily Light GBM

It achieved a mean accuracy score of 61% in the validation and 62% in the testing dataset. An MCC of 0.2793 indicates that the model is not random and has predictive power. The *p*-value of 0.0049 indicates that the MCC is statistically significant at a 2.5% significance level. It is observed from Figure 3 and Table A1 that the one-day prior change in the value of the VIX Index ranked the highest and, subsequently, the two-day, three-day, and four-day prior change in the value of the VIX ranked third, fifth, and sixth, respectively. Hence, it can be stated that the VIX's closing value is the most important for predicting the

next day's labels in sequence. Additionally, the gold price, the Economic Policy Uncertainty Index: Infectious Disease Tracker (EPU IDT Index), the U.S. Dollar Index, the CBOE Crude Oil ETF Volatility Index, and the TED Spread are ranked significantly in the top ten. It is further observed that the economic policy uncertainty indices ranked multiple times in the ranked features. Among the fixed income securities, the TED Spread, Credit Spread, option-adjusted spread, the 10-year yield on inflation-indexed, and the Fed Rate do make an impact, but the TED Spread and the Credit Spread are more significant.

Hence, it can be stated that economic policy uncertainty indices, gold price, USD Index, CBOE Crude Oil ETF Volatility Index, and TED Spread are termed as strong predictors. Most of them are ranked under the top ten features and repeated under a ranked set of twenty-nine features. Additionally, 10-year yield on inflation-indexed instruments, option-adjusted spread, interest rate, and the spread on bonds do also make an impact on predicting the daily movement of the VIX Index. The redundant features are the 1M T-Yield Curve Rate, 4w T-Bill Rate, 5Y Breakeven Inflation Rate, 5Y Yield Inflation Indexed, AAA Corp OAS, BBB Corp OAS, BBB Corp Yield, Corp Yield, Silver Price, and Yield Spread.

*4.2. Results from Daily XG Boost*

It achieved a mean accuracy score of 55.62% in the validation and 62% in the testing datasets. The MCC of 0.2793 indicates that the model is not random and has predictive power. The *p*-value of 0.0049 signifies that the MCC is statistically significant. It can be revealed from Figure 5 and Table A2 that the changes in the VIX Index are significant for predicting its movements. The changes in the Economic Policy Uncertainty Index: Infectious Disease Tracker (EPU IDT Index) are the most important feature. Other important features are the Gold Price, USD Index, EPU Policy and Equity Index, Silver Price, Fed Rate, and the TED Spread. The AAA Corp OAS, 5Y Breakeven Inflation Rate, OVX, and the 4w T-Bill Rate are also important, but they are repeated once in the top twenty-two important features. Redundant features are the 10Y Breakeven Inflation Rate, 10Y Yield Inflation Indexed, 1M T-Yield Curve Rate, 5Y Yield Inflation Indexed, AAA Corp Yield, BBB Corp OAS, BBB Corp Yield, Corp OAS, Corp Yield, Credit Spread, and Yield Spread.

*4.3. Results from Daily Logistic Regression*

It achieved a mean accuracy score of 47% in the validation and 52% in the testing datasets. The MCC of 0.0162 and *p*-value of 0.8733 indicate that it is a weak classifier, and the result is not significant. Fortunately, it is only used here for anticipating the directional relationship. It is found from Figure 7 and Table A3 that the changes in the VIX Index are the most important, and their negative coefficients could roughly indicate a reversal tendency. Economic policy uncertainty indices, Gold Price, U.S. Dollar Index, Silver Price, and TED Spread are strong predictors because their coefficients are comparatively highly sensitive. Since all the coefficients of the Economic Policy Uncertainty Index: Infectious Disease Tracker (EPU IDT Index) are positive, it indicates a positive association with the VIX Index, while Economic Policy Uncertainty Equity Market Index is roughly negatively associated with the VIX Index. However, the Economic Policy Uncertainty Index is somewhat positively related. Additionally, the yield on inflation-indexed instruments, the 1-month Treasury yield curve rate, the 4-week Treasury bill rate, and the other spreads do also make an impact on the VIX Index, but fixed income securities are relatively low predictors. The all-positive coefficients of the credit spread indicate that the VIX Index increases when the credit spread increases. Inflation is positively related to the VIX Index, while the CBOE Crude Oil ETF Volatility Index is negatively associated. Because a higher credit spread and inflation are the negative signals for the economy, they are positively associated with the VIX Index.

Furthermore, the coefficients of 1-day, 2-day, 3-day, 4-day, and 5-day prior returns on the gold price are positive, negative, positive, negative, and positive, respectively. This could indicate that the level of the VIX Index is positively associated with the previous day's return on the gold price and negatively associated with a day-before-yesterday's

return on the gold price. Hence, it can be loosely speculated that the VIX Index and gold price are concurrently negatively associated.

### 4.4. Results from Weekly Light GBM

It achieved a mean accuracy score of 62% in the validation and 62% in the testing datasets. The MCC of 0.2522 indicates that the model is not random and has predictive power. The *p*-value of 0.0772 indicates that the MCC is not significant at a 2.5% significant level; however, it can be significant at 7.72% or a higher significance level. The weekly model does not do well as compared to the daily model because the VIX Index tried to subside itself during a week's time. From Figure 4 and Table A1, the previous week's change in the value of the VIX Index is the most significant. The Financial Stress Index is a strong predictor. The Inflation rate, TED Spread, Fed Rate, M2 Money Supply, initial unemployment claims, U.S. Dollar Index, Gold Price, Crude Oil Price, and CBOE Crude Oil ETF Volatility Index, in addition to the spread on fixed income securities, affect the VIX Index significantly but with lesser intensity. The Economic Policy Uncertainty Equity Market Index does qualify in the ranked features, but the Economic Policy Uncertainty: Infectious Disease Tracker (EPU IDT Index) does not because it can be speculated that EPU IDT Index goes up or down and returns to the level in a week's time. The list of redundant features is the 10Y Yield Inflation Indexed, 1m T-Yield Curve Rate, 30Y Mortgage Rate, 3m T-Bill 2nd Rate, 4w T-Bill 2nd Rate, 4w T-Bill Rate, Bank Prime Loan Rate, C EPU IDT Index, Corp OAS, and R Silver Price.

### 4.5. Results from Weekly XG Boost

It achieved a mean accuracy score of 57.81% in the validation and 60% in the testing datasets. The MCC of 0.2077 indicates that the model is not random and has some predictive power. However, the *p*-value of 0.1477 indicates that the MCC is not statistically significant. The performance is even poorer to that of the weekly Light GBM. It gives importance to only three features, the 5Y Breakeven Inflation Rate and the Financial Stress Index in addition to the VIX Index. It sets rest of the features' scores to zero.

### 4.6. Results from Weekly Logistic Regression

It achieved a mean accuracy score of 56% in the validation and 44% in the testing datasets. The MCC of $-0.0933$ indicates that the classifier is useless and the *p*-value of 0.5192 further indicates the weaker predictability of the classifier. The previous week's change in the VIX Index is the most significant, as shown in Figure 6 and Table A3. The Economic Policy Uncertainty Equity Market Index, AAA-rated corporate bond option-adjusted spread, TED Spread, and the inflation rate are strong predictors and positively correlated. Surprisingly, the Credit Spread, inflation-indexed bonds, and initial claims (NSA) on unemployment are negatively, but significantly, associated with the VIX Index. Some findings are contradicted with the daily logistic regression model.

Considering the performance of all six models applied in this study, as the Light GBM and XG Boost outperformed the logistic regression, the decision made by the Light GBM and XG Boost supersedes the logistic regression. The performances of the daily Light GBM and XG Boost models are comparable, while the performance of the weekly XG Boost deteriorated compared to the performance of the weekly Light GBM, though both performed poorly. As a weak classifier, logistic regression is applied in this study for the estimation of the directional relationship. It can be inferred from the study that the economic policy uncertainty indices, gold price, USD Index, CBOE Crude Oil ETF Volatility Index, Financial Stress Index, and the yield on inflation-indexed Treasury securities are termed as strong predictors. Moreover, the credit spread and TED spread do make a significant impact on the VIX Index, but the impact of the other fixed-income securities is placed thereafter. The M2 money supply, the initial claims on unemployment, and Fed rate are also strong predictors. The weekly Light GBM has given importance to the yield spread.

Both the Economic Policy Uncertainty Index of Equity Market Volatility: Infectious Disease Tracker (EPU IDT Index) and Economic Policy Uncertainty Daily Policy Index (EPU Index) are positively associated with the level of the VIX Index, and this relationship is in line with the findings of Su et al. (2019), but the findings from our research are more comprehensive. However, the Economic Policy Uncertainty Equity Market Index is mostly negatively associated with the level of the VIX Index. Some have given importance to 5Y Yield Inflation Indexed, while some have given importance to 10Y Yield Inflation Indexed.

The features that are weak to redundant are the 1M T-Yield Curve Rate, 5Y Yield Inflation Indexed, 10Y Yield Inflation Indexed, BBB Corp OAS, BBB Corp Yield, Corp Yield, Silver Price, Yield Spread, and 4w T-Bill Rate, though some have conflicting importance.

## 5. Conclusions

As predicting the day-to-day and week-to-week movements of the VIX Index is interesting, and its association with the macroeconomic variables is highly important, machine learning algorithms, such as logistic regression, XG Boost, and Light GBM, are applied on the set of feature variables derived from the daily and weekly macroeconomic variables and a closing value of the VIX Index, and after performing hyperparameters tuning, the captured feature variables are ranked according to their importance in predicting the daily and weekly movements of the VIX Index. Thereafter, the models are asked to predict the binary labels in the testing datasets. The outcome of the models was depicted and analyzed in detail in Section 4.

It can be suggested from the results that, with the given set of feature variables, the Light GBM achieved an accuracy score of 62% for both the daily and weekly models, which is higher than that of the logistic regression. The Daily XG Boost model was also accurate. However, when the t-test on the MCC is taken into consideration, the performances of the weekly models based on their *p*-values deteriorated, though the weekly Light GBM performed slightly better. Hence, the decision made by the daily Light GBM and XG Boost and the weekly Light GBM on the ranking features can be significantly accepted. The economic policy uncertainty indices, gold price, USD Index, and CBOE Crude Oil ETF Volatility Index are termed as strong predictors. The Financial Stress Index, Treasury securities, M2 money supply, initial claims on unemployment, Fed rate, credit spread, and the TED spread are also strong predictors, while various yields on fixed income securities make a little less impact on the VIX Index. The Financial Stress Index and the TED spread are positively related to the VIX Index, while the credit spread and the yield spread have conflicting results in a directional relationship with the VIX Index. The features that are weak to redundant are the 1M T-Yield Curve Rate, yield on inflation-indexed security, BBB Corp OAS, BBB Corp Yield, Corp Yield, Silver Price, Yield Spread, and the 4w T-Bill Rate, though some have conflicting importance levels.

Both the Economic Policy Uncertainty Index of Equity Market Volatility: Infectious Disease Tracker and the Economic Policy Uncertainty Daily Policy Index (EPU Index) are positively associated with the level of the VIX Index, and this relationship is in line with the findings of Su et al. (2019). However, the Economic Policy Uncertainty Equity Market Index is mostly negatively associated with the level of the VIX Index.

**Author Contributions:** Data curation, formal analysis, investigation, methodology, software, writing—original draft, validation, visualization, Writing—review and editing, A.P.; Formal analysis, investigation, resources, writing—review and editing, validation, P.B.; Formal analysis, investigation, conceptualization, supervision, validation, A.S. All authors have read and agreed to the published version of the manuscript.

**Funding:** This research received no external funding.

**Institutional Review Board Statement:** Not applicable.

**Informed Consent Statement:** Not applicable.

**Data Availability Statement:** The authors have declared that this research is based on publicly available data.

**Conflicts of Interest:** The authors have declared that there is no conflict of interest for this article.

**Practical Implications:** The findings of this research are important for traders and investors in anticipating the risk in the U.S. stock market. It is suggested that they should take the decision made by Light GBM on ranking features, as these results are more significant compared to the logistic regression. The decision made by the XG Boost can also be taken into consideration, as its performance is comparable to that of the Light GBM. Moreover, as the performance of the stock market in other countries is highly influenced by the performance of the U.S. stock market, so this study is useful for traders and investors of other countries as well. The outcome of this study is useful for policymakers as, based on Volatility Index and macro variables, they can make effective policies.

**Research Limitations and Future Scope:** This investigation is limited to the U.S. market and a few techniques. Other ensemble learning algorithms such as random forest, histogram-based gradient boosting, and extreme gradient boosting, along with principal component analysis, needed to rank the set of feature variables can also be considered along with a wide-ranging set of macro variables at a monthly level in the future on the U.S. market, as well as on the data of other economies.

## Appendix A

The ranked features, along with their scores from the Light GBM and XG Boost models, are displayed in Tables A1 and A2, respectively, and the ranked features along with their feature coefficients from the daily and weekly logistic regression models are stated in Tables A3 and A4, respectively.

**Table A1.** Complete list of ranked features and their score from Light GBM models. Numbers are truncated to four decimal places.

| Daily | | | Weekly | | |
|---|---|---|---|---|---|
| Rank | Feature Names | Feature Scores | Rank | Feature Names | Feature Scores |
| 1 | C VIX-1 | 11.6069 | 1 | C VIX-1 | 0.2285 |
| 2 | R Gold Price-3 | 7.5021 | 2 | Financial Stress Index-1 | 0.0858 |
| 3 | C VIX-2 | 4.8436 | 3 | 10Y Breakeven Inflation Rate-1 | 0.0633 |
| 4 | C EPU IDT Index-5 | 3.5566 | 4 | TED Spread-1 | 0.0628 |
| 5 | C VIX-3 | 3.3592 | 5 | Fed Rate-1 | 0.0564 |
| 6 | C VIX-4 | 3.3311 | 6 | M2 Money Supply-1 | 0.0555 |
| 7 | R USD Index-4 | 1.9900 | 7 | R Initial Claims SA-1 | 0.0513 |
| 8 | C OVX-1 | 1.2725 | 8 | 5Y Breakeven Inflation Rate-1 | 0.0382 |
| 9 | TED Spread-5 | 1.1905 | 9 | Corp Yield-1 | 0.0354 |
| 10 | C OVX-5 | 1.1830 | 10 | Yield Spread-1 | 0.0312 |
| 11 | C EPU Equity Index-5 | 0.9776 | 11 | BBB Corp OAS-1 | 0.0271 |
| 12 | Credit Spread-5 | 0.7099 | 12 | AAA Corp OAS-1 | 0.0254 |
| 13 | C EPU Equity Index-3 | 0.7088 | 13 | C OVX-1 | 0.0239 |
| 14 | C EPU IDT Index-1 | 0.6420 | 14 | R USD Index-1 | 0.0217 |
| 15 | C EPU Equity Index-2 | 0.5535 | 15 | R Gold Price-1 | 0.0201 |
| 16 | Fed Rate-4 | 0.4905 | 16 | AAA Corp Yield-1 | 0.0140 |
| 17 | C OVX-2 | 0.4797 | 17 | R Crude Oil Prices-1 | 0.0133 |
| 18 | AAA Corp Yield-1 | 0.3643 | 18 | BBB Corp Yield-1 | 0.0079 |

**Table A1.** *Cont.*

| | Daily | | | Weekly | |
|---|---|---|---|---|---|
| **Rank** | **Feature Names** | **Feature Scores** | **Rank** | **Feature Names** | **Feature Scores** |
| 19 | C EPU Index-5 | 0.2715 | 19 | Credit Spread-1 | 0.0061 |
| 20 | 10Y Yield Inflation Indexed-1 | 0.2630 | 20 | C EPU Equity Index-1 | 0.0040 |
| 21 | 10Y Breakeven Inflation Rate-1 | 0.2519 | 21 | R Initial Claims NSA-1 | 0.0023 |
| 22 | R Gold Price-5 | 0.1657 | 22 | 5Y Yield Inflation Indexed-1 | 0.0015 |
| 23 | 10Y Breakeven Inflation Rate-5 | 0.1176 | 23 | C EPU Index-1 | 0.0013 |
| 24 | 10Y Yield Inflation Indexed-4 | 0.0745 | | | |
| 25 | Corp OAS-1 | 0.0707 | | | |
| 26 | R Gold Price-4 | 0.0284 | | | |
| 27 | TED Spread-2 | 0.0148 | | | |
| 28 | TED Spread-4 | 0.0130 | | | |
| 29 | Corp OAS-4 | 0.0006 | | | |

**Table A2.** Complete list of ranked features and their score from XG Boost. Numbers are truncated to four decimal places.

| | Daily | | | Weekly | |
|---|---|---|---|---|---|
| **Rank** | **Feature Names** | **Feature Scores** | **Rank** | **Feature Names** | **Feature Scores** |
| 1 | C EPU IDT Index-1 | 0.0586 | 1 | C VIX-1 | 0.3464 |
| 2 | R Gold Price-4 | 0.0568 | 2 | 5Y Breakeven Inflation Rate-1 | 0.3305 |
| 3 | C VIX-1 | 0.0544 | 3 | Financial Stress Index-1 | 0.3231 |
| 4 | C EPU IDT Index-5 | 0.0539 | | | |
| 5 | C EPU IDT Index-3 | 0.0520 | | | |
| 6 | R USD Index-3 | 0.0513 | | | |
| 7 | C VIX-4 | 0.0494 | | | |
| 8 | C EPU Index-4 | 0.0490 | | | |
| 9 | AAA Corp OAS-1 | 0.0475 | | | |
| 10 | 5Y Breakeven Inflation Rate-5 | 0.0475 | | | |
| 11 | C VIX-2 | 0.0468 | | | |
| 12 | C OVX-5 | 0.0463 | | | |
| 13 | R Silver Price-4 | 0.0440 | | | |
| 14 | Fed Rate-5 | 0.0434 | | | |
| 15 | R Gold Price-3 | 0.0431 | | | |
| 16 | R Silver Price-5 | 0.0418 | | | |
| 17 | TED Spread-4 | 0.0416 | | | |
| 18 | C EPU Index-1 | 0.0374 | | | |
| 19 | C EPU Index-5 | 0.0365 | | | |
| 20 | C EPU Equity Index-1 | 0.0352 | | | |
| 21 | R USD Index-5 | 0.0332 | | | |
| 22 | 4w T-Bill Rate-3 | 0.0300 | | | |

**Table A3.** Complete list of daily ranked features and their coefficient from Logistic Regression. Numbers are truncated to six decimal places.

| Rank | Feature Names | Feature Coefficients | Rank | Feature Names | Feature Coefficients |
|---|---|---|---|---|---|
| 1 | C VIX-1 | −0.016567 | 61 | Corp Yield-5 | −0.001451 |
| 2 | C VIX-4 | −0.009427 | 62 | BBB Corp Yield-5 | −0.001405 |
| 3 | C EPU Equity Index-1 | −0.006827 | 63 | AAA Corp Yield-1 | 0.001356 |
| 4 | C EPU Index-1 | 0.006763 | 64 | C EPU Equity Index-2 | −0.001347 |
| 5 | R Gold Price-4 | −0.006366 | 65 | Credit Spread-3 | 0.001347 |
| 6 | C EPU IDT Index-3 | 0.005432 | 66 | Fed Rate-3 | −0.001234 |
| 7 | C VIX-5 | −0.005105 | 67 | C OVX-2 | −0.001233 |
| 8 | R Gold Price-2 | −0.005022 | 68 | Credit Spread-5 | 0.001219 |
| 9 | C EPU IDT Index-5 | 0.004941 | 69 | 5Y Breakeven Inflation Rate-4 | −0.001209 |
| 10 | R Gold Price-3 | 0.004702 | 70 | BBB Corp Yield-2 | −0.001155 |
| 11 | C VIX-2 | −0.004614 | 71 | Credit Spread-1 | 0.001144 |
| 12 | R Silver Price-4 | 0.004530 | 72 | C EPU Index-3 | 0.001140 |
| 13 | R USD Index-5 | −0.004159 | 73 | BBB Corp OAS-5 | −0.001124 |
| 14 | TED Spread-3 | 0.004091 | 74 | TED Spread-1 | −0.001105 |
| 15 | 5Y Yield Inflation Indexed-1 | 0.003994 | 75 | AAA Corp Yield-5 | −0.001104 |
| 16 | 10Y Breakeven Inflation Rate-1 | 0.003698 | 76 | Corp OAS-3 | −0.001055 |
| 17 | R Silver Price-5 | 0.003649 | 77 | 4w T-Bill Rate-3 | −0.001001 |
| 18 | R USD Index-4 | 0.003626 | 78 | Fed Rate-1 | −0.000998 |
| 19 | 1M T-Yield Curve Rate-1 | 0.003505 | 79 | 1M T-Yield Curve Rate-3 | −0.000935 |
| 20 | R Gold Price-5 | 0.003496 | 80 | Corp OAS-4 | −0.000933 |
| 21 | AAA Corp OAS-5 | 0.003361 | 81 | Corp OAS-2 | −0.000926 |
| 22 | BBB Corp OAS-1 | −0.003303 | 82 | R USD Index-1 | −0.000922 |
| 23 | 4w T-Bill Rate-1 | 0.003218 | 83 | 5Y Yield Inflation Indexed-4 | 0.000888 |
| 24 | C EPU IDT Index-4 | 0.003151 | 84 | 10Y Breakeven Inflation Rate-5 | −0.000866 |
| 25 | AAA Corp OAS-2 | 0.003104 | 85 | 10Y Yield Inflation Indexed-2 | −0.000866 |
| 26 | 10Y Breakeven Inflation Rate-3 | 0.002965 | 86 | 10Y Yield Inflation Indexed-4 | 0.000826 |
| 27 | C EPU Equity Index-4 | −0.002910 | 87 | R Silver Price-2 | 0.000762 |
| 28 | AAA Corp OAS-1 | 0.002896 | 88 | Corp OAS-5 | −0.000743 |
| 29 | 10Y Breakeven Inflation Rate-2 | 0.002781 | 89 | 5Y Breakeven Inflation Rate-2 | −0.000735 |
| 30 | C VIX-3 | −0.002749 | 90 | BBB Corp Yield-3 | −0.000731 |
| 31 | 5Y Breakeven Inflation Rate-5 | −0.002674 | 91 | Yield Spread-2 | −0.000715 |
| 32 | C EPU IDT Index-2 | 0.002657 | 92 | Yield Spread-4 | 0.000629 |
| 33 | R Silver Price-3 | 0.002651 | 93 | 5Y Breakeven Inflation Rate-3 | −0.000614 |
| 34 | AAA Corp OAS-3 | 0.002649 | 94 | Corp Yield-1 | 0.000586 |
| 35 | C EPU Index-4 | 0.002640 | 95 | C EPU Index-2 | −0.000579 |
| 36 | C EPU Equity Index-3 | 0.002583 | 96 | AAA Corp Yield-2 | −0.000571 |
| 37 | R Gold Price-1 | 0.002523 | 97 | Fed Rate-4 | −0.000506 |
| 38 | TED Spread-5 | 0.002439 | 98 | 5Y Yield Inflation Indexed-5 | 0.000477 |
| 39 | Yield Spread-3 | 0.002292 | 99 | 5Y Yield Inflation Indexed-2 | 0.000472 |
| 40 | Corp OAS-1 | −0.002287 | 100 | Corp Yield-2 | −0.000466 |
| 41 | C EPU IDT Index-1 | 0.002242 | 101 | R USD Index-3 | −0.000454 |
| 42 | 4w T-Bill Rate-5 | −0.002179 | 102 | 1M T-Yield Curve Rate-2 | 0.000447 |
| 43 | 5Y Breakeven Inflation Rate-1 | −0.002133 | 103 | 4w T-Bill Rate-4 | 0.000440 |

**Table A3.** *Cont.*

| Rank | Feature Names | Feature Coefficients | Rank | Feature Names | Feature Coefficients |
|------|---------------|----------------------|------|---------------|----------------------|
| 44 | C OVX-4 | −0.002116 | 104 | R USD Index-2 | −0.000415 |
| 45 | Yield Spread-5 | −0.002078 | 105 | Yield Spread-1 | −0.000408 |
| 46 | BBB Corp OAS-2 | −0.002023 | 106 | AAA Corp Yield-3 | −0.000405 |
| 47 | AAA Corp OAS-4 | 0.001993 | 107 | Credit Spread-4 | 0.000403 |
| 48 | BBB Corp OAS-3 | −0.001993 | 108 | AAA Corp Yield-4 | −0.000333 |
| 49 | TED Spread-4 | 0.001972 | 109 | Fed Rate-2 | 0.000284 |
| 50 | Fed Rate-5 | −0.001895 | 110 | BBB Corp Yield-1 | −0.000280 |
| 51 | 1M T-Yield Curve Rate-5 | −0.001881 | 111 | C OVX-3 | −0.000256 |
| 52 | Credit Spread-2 | 0.001861 | 112 | BBB Corp Yield-4 | −0.000243 |
| 53 | 10Y Yield Inflation Indexed-1 | 0.001811 | 113 | Corp Yield-3 | −0.000208 |
| 54 | 10Y Breakeven Inflation Rate-4 | 0.001705 | 114 | R Silver Price-1 | −0.000206 |
| 55 | BBB Corp OAS-4 | −0.001651 | 115 | 4w T-Bill Rate-2 | 0.000146 |
| 56 | C OVX-5 | −0.001575 | 116 | 5Y Yield Inflation Indexed-3 | −0.000067 |
| 57 | TED Spread-2 | 0.001534 | 117 | C EPU Index-5 | −0.000059 |
| 58 | C OVX-1 | −0.001501 | 118 | 1M T-Yield Curve Rate-4 | 0.000046 |
| 59 | C EPU Equity Index-5 | −0.001490 | 119 | 10Y Yield Inflation Indexed-3 | 0.000023 |
| 60 | 10Y Yield Inflation Indexed-5 | −0.001490 | 120 | Corp Yield-4 | −0.000001 |

**Table A4.** Complete list of weekly ranked features and their coefficient from Logistic Regression. Numbers are truncated to four decimal places.

| Rank | Feature Names | Feature Coefficients | Rank | Feature Names | Feature Coefficients |
|------|---------------|----------------------|------|---------------|----------------------|
| 1 | C VIX-1 | −0.012452 | 18 | Corp OAS-1 | −0.001903 |
| 2 | C EPU Index-1 | 0.008386 | 19 | C EPU Equity Index-1 | −0.001515 |
| 3 | AAA Corp OAS-1 | 0.007732 | 20 | Corp Yield-1 | −0.001426 |
| 4 | TED Spread-1 | 0.006576 | 21 | Fed Rate-1 | 0.001374 |
| 5 | 10Y Breakeven Inflation Rate-1 | 0.006219 | 22 | C OVX-1 | 0.001180 |
| 6 | Credit Spread-1 | −0.005811 | 23 | 5Y Yield Inflation Indexed-1 | 0.001029 |
| 7 | BBB Corp OAS-1 | −0.005482 | 24 | Bank Prime Loan Rate-1 | 0.001023 |
| 8 | 10Y Yield Inflation Indexed-1 | −0.005228 | 25 | Yield Spread-1 | 0.000774 |
| 9 | R Initial Claims NSA-1 | −0.005188 | 26 | R USD Index-1 | −0.000649 |
| 10 | AAA Corp Yield-1 | 0.005147 | 27 | M2 Money Supply-1 | 0.000577 |
| 11 | BBB Corp Yield-1 | −0.004549 | 28 | 30Y Mortgage Rate-1 | 0.000369 |
| 12 | R Crude Oil Prices-1 | −0.003733 | 29 | C EPU IDT Index-1 | 0.000341 |
| 13 | R Gold Price-1 | 0.003600 | 30 | R Initial Claims SA-1 | 0.000331 |
| 14 | 4w T-Bill 2nd Rate-1 | −0.003268 | 31 | R Silver Price-1 | −0.000299 |
| 15 | 3m T-Bill 2nd Rate-1 | -0.002975 | 32 | 1m T-Yield Curve Rate-1 | −0.000267 |
| 16 | 5Y Breakeven Inflation Rate-1 | −0.002360 | 33 | 4w T-Bill Rate-1 | −0.000210 |
| 17 | Financial Stress Index-1 | −0.002215 | | | |

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
