# Peer review of "The Impact of the U.S. Macroeconomic Variables on the CBOE VIX Index"

_jrfm, doi:10.3390/jrfm15030126_

Round 1
Reviewer 1 Report
- Typos need to be corrected e.g., “..(HAR..”, or terminology like “ ..by a set of matrixes: accuracy score, …”, etc.
- Tables 5, 6 and 7: numbers are in the form of “,00”. Is this the actual result of a rounding to the closest integer?
- The authors compare the performance of two different classification algorithms while in the 1st reference of the paper a wide variety of algorithms is being considered for the same problem. To support the conclusion that Light GBM is the best performing algorithm for the selected parameters and the analyzed sample of data some additional algorithms could be included so as to make the paper conclusions stronger.
- The paper provides a wide range of parameters for the classification problem analyzed. Looking on the importance of these parameters, it appears that there may be parameters with a marginal role in the process of predicting VIX. Therefore it would be interesting to have a view on the performance of the same algorithms when the parameters are limited based on their importance.
Author Response
1: Noted and rectified
2: They are truncated to four decimal places or two decimal places in percent form. I now provided better explanation in the manuscript.
3: I have also included XG Boost for the same purpose. The daily XG Boost model performed well. Further details can be seen in the manuscript.
4: The focus of the research is to find the importance of the feature variables from the list of available features. However, I now tried to indicate features that are weak to redundant.
Reviewer 2 Report
The current manuscript is written and presented with a lot of details in the research steps and results. Some minor points are required to improve or clarify.
- There is no research hypothesis constructed and empirically tested in this paper. It will be more rigorous if research hypotheses are constructed from theories and/or existing literature.
- Please mentioned in the paper the reasons for choosing equal periods for training, testing, validation (100 day and 50 weeks)?
- Conclusion part seems too weak. As mentioned, research objectives clarified could help here with emphasizing most important implications and revealing the contribution of this research to the scientific knowledge.
4. The paper does not respect the template of the review especial in the presentations of tables and references.
Author Response
1: The motive of this research is to anticipate the importance of the macroeconomic variables while predicting the movement of the CBOE VIX Index on next day by setting up the problem in classification techniques. I further applied t-test on Matthews correlation coefficient as you suggested. It is incorporated in Section 3.6.
2: Figure 2 depict that the length of the testing datasets and the validation datasets are 100 days for daily models and 50 weeks for the weekly models. Length of the training datasets is not 100 days or neither 50 weeks but is all the preceding data prior to the validation segment. A machine learning model requires periodic hyper-tuning, because predicting a faraway in the future would not be a good idea. Hence, a fixed and sufficient window of the validation and testing datasets would be advisable. It is incorporated in Section 3.7.
3: Noted and rectified
4: Noted and tried to rectify as per my understanding